# Multiple-UAV Reinforcement Learning Algorithm Based on Improved PPO in Ray Framework

**Guang Zhan** [1,*] **, Xinmiao Zhang** [2] **, Zhongchao Li** [3] **, Lin Xu** [2] **, Deyun Zhou** [1] **and Zhen Yang** [1]

[1] School of Electronics and Information, Northwestern Polytechnical University, Xi'an 710072, China; dyzhou@nwpu.edu.cn (D.Z.); bhuyz@buaa.edu.cn (Z.Y.)

[2] College of Information Science and Engineering, Northeastern University, Shenyang 110004, China; 1900934@stu.neu.edu.cn (X.Z.); xulin@mail.neu.edu.cn (L.X.)

[3] Shenyang Aircraft Design and Research Institute, Aviation Industry Corporation of China, Shenyang 110035, China; m13940334601@163.com

[*] Correspondence: zhanguang@mail.nwpu.edu.cn

**Abstract:** Distributed multi-agent collaborative decision-making technology is the key to general artificial intelligence. This paper takes the self-developed Unity3D collaborative combat environment as the test scenario, setting a task that requires heterogeneous unmanned aerial vehicles (UAVs) to perform a distributed decision-making and complete cooperation task. Aiming at the problem of the traditional proximal policy optimization (PPO) algorithm's poor performance in the field of complex multi-agent collaboration scenarios based on the distributed training framework Ray, the Critic network in the PPO algorithm is improved to learn a centralized value function, and the muti-agent proximal policy optimization (MAPPO) algorithm is proposed. At the same time, the inheritance training method based on course learning is adopted to improve the generalization performance of the algorithm. In the experiment, MAPPO can obtain the highest average accumulate reward compared with other algorithms and can complete the task goal with the fewest steps after convergence, which fully demonstrates that the MAPPO algorithm outperforms the state-of-the-art.

**Keywords:** multiple UAVs; deep reinforcement learning; PPO; curriculum learning; Ray

## 1. Introduction

Multiple unmanned aerial vehicle (UAV)-coordinated air combat refers to a mode of warfare in which two or more UAVs cooperate with each other to complete air combat tasks, including coordinated mobility, coordinated strikes, and fire cover, which is a concrete embodiment of the modern integrated naval, land, air, space, and electrical operation mode in multi-aircraft air combat. Therefore, improving the coordination efficiency of multiple UAVs is of great significance to obtain air supremacy on the battlefield and reduce combat casualties. However, compared with the air combat decision making of a single UAV, the multiple UAV coordination problem involves a series of challenges such as intelligent decision-making [1], distributed collaboration [2], and formation management [3].

Supervised learning is a task-driven learning method. The quality and quantity of labeled samples determine the ceiling of its ability. However, preprocessing labeled samples and data is a time-consuming and laboring task. Unlike supervised learning, the training data of reinforcement learning do not require labels, which are completely obtained by the behavior of the agent. There is only a reward function throughout the learning process. The research problem of reinforcement learning is how the agent learns a certain policy through interaction with the environment to maximize the accumulated reward [4].

As a non-labeled decision learning mechanism, reinforcement learning can make up for the deficiencies of supervised learning to a certain extent. With the improvement of hardware computing power and the proposal of many efficient distributed training frameworks, deep reinforcement learning (DRL), which combines the excellent perception ability of deep learning and the decision-making ability of reinforcement learning, has

walked into people's vision. DRL realizes end-to-end learning from the original input of data to decision making and has been widely used in a series of problems such as unmanned driving [5], resource scheduling [6,7], and navigation [8,9]. In the field of UAV decision-making, Syed et al. [10] designed a novel control and testing platform based on Q-learning for a smart morphing wing system that was introduced to obtain optimal aerodynamic properties. Zhou et al. [11] proposed a fast adaptive fault estimator-based active fault-tolerant control strategy for a quadrotor UAV against multiple actuator faults. Zhang et al. [12] presented an improved deep deterministic policy gradient (DDPG) algorithm for the path-following control problem of UAVs and designed a specific reward function to minimize the cross-track error of the path-following problem.

However, real intelligence includes the wisdom of groups. Compared with a single agent, the cooperation between agents has broader application scenarios. However, multi-agent deep reinforcement learning (MADRL) faces many challenges. The first challenge is the non-static nature of the environment. In the multi-agent system, agents update their own policies independently. From the perspective of any agent, agents are in an unstable environment, and the non-static nature of the environment violates the Markov hypothesis. The second challenge is that, with increasing environmental complexity, the traditional policy gradient algorithm has high variance estimation [13]. The third challenge is that, with an increase in the number of agents, the exploration space of agents increases exponentially, which requires great computing resources.

To handle the impacts from system uncertainties and a dynamic environment, Jafari et al. [14] designed a novel reinforcement learning technique, which is appropriate for real-time implementation and has been integrated with multi-agent flocking control. Liu et al. [15] proposed a distributed model-free solution based on reinforcement learning for the leader–follower formation control problem of heterogeneous multi-agent systems. Lowe et al. [16] presented a method of "centralized training and decentralized execution", which performs well in multiple task scenarios such as multi-agent competition and cooperation. Li et al. [17] improved the multi-agent deep deterministic policy gradient (MADDPG) algorithm when solving multi-agent decision-making problems and adopted the method of transfer learning to improve the generalization performance of the algorithm. MADDPG uses a deterministic policy to interact with the environment, which is not conducive to the exploration of actions, and it is easy for the agent to fall into the local optimal solution. In order to train random policies, Schulman et al. [18] proposed a proximal policy optimization (PPO) algorithm. On the one hand, it avoided complex KL divergence calculations, but on the other hand, it turned the constrained optimization problem into an unconstrained optimization problem and obtained similar performance to the original trust region policy optimization (TRPO) [19]. Now, PPO is the mainstream algorithm in the field of reinforcement learning. Hoseini et al. [20] applied the PPO algorithm to the design of airborne battery power; this increased the flight time of the UAV, which is a major breakthrough of PPO algorithm in the field of UAV. However, the original PPO algorithm was proposed for a single-agent environment and performs poorly in multi-agent environments. At present, the research of applying the PPO algorithm to the cooperation of multiple UAVs is not very sufficient.

In this work, we use the idea of centralized training and distributed execution to make the Critic network learn a global value function and extend the PPO algorithm to the field of multi-agent. Under the distributed training framework Ray [21], we propose the multi-agent proximal policy optimization (MAPPO) algorithm to explore how to learn the optimal joint behavior between heterogeneous UAVs in a self-developed Unity3D UAV collaborative combat environment. The main innovations of this paper are as follows:

1. MAPPO uses the Critic network with global information and the Actor network with local information to achieve cooperation between heterogeneous UAVs, and the action entropy reward is added to the objective function to encourage the exploration of UAVs;
2. The policy network of homogeneous UAVs realizes parameter sharing, and each UAV has the ability to make independent decisions;

3. A staged training method based on course learning is proposed to improve the generalization of the algorithm.

Experiments are conducted in our challenging multi-agent collaborative environment with a high-dimensional state space. Results show that MAPPO outperforms state-of-the-art methods and demonstrates its potential value for large-scale real-world applications.

The remainder of this manuscript is structured as follows. Section 2 introduces some background knowledge of Markov Game and PPO algorithms. Section 3 formulates the the overall framework of the MAPPO algorithm. Section 4 elaborates on the modeling process of the collaborative combat environment. An ablation experiment and comparative experiment are presented in Section 5. Section 6 concludes this paper and envisages some future work.

## 2. Background

### 2.1. Markov Game

Single-agent reinforcement learning is described by the Markov decision process, while multi-agent reinforcement learning needs to be described as a Markov game [22]. Markov games are also called stochastic games. In multi-agent environments, Markov games are composed of the following elements:

1. The global states of all agents is represented by $S$;
2. $a_1 \ldots a_n$ indicates the action of each agent;
3. $s_1 \ldots s_n$ represents the observation of each agent.

In the process of a Markov game, the initial state can be determined by a random distribution. The goal of all reinforcement learning algorithms is to learn an optimal mapping function $\pi$ from state to action to maximize the accumulated reward obtained by the agents. The objective function to be optimized is:

$$R_i = \sum_{t=0}^{T} \gamma^t r_i^t \tag{1}$$

where $\gamma$ is a discount factor, indicating the effect of future rewards on current agent behavior. $\gamma = 1$ means that the reward value of the future state has a great influence on the action-state function, while $\gamma = 0$ means that the reward value of the future state has little influence on the action-state function. $r$ is the reward function, representing the reward obtained by the agent after executing action $a$ at state $s$.

### 2.2. PPO Algorithm

PPO algorithm is a new policy gradient algorithm. In the policy gradient algorithm, the agent updates the policy by gradient boosting. Although the traditional policy gradient algorithms, such as advantage actor critic (A2C) [23] and actor critic with experience replay (Acer) [24], have achieved good control effects in many decision-making problems, they still face many problems, such as the difficult selection of iteration step size and low data utilization.

Figure 1 describes the training flow chart of the PPO algorithm. During training, a batch of samples are selected from the buffer to update network parameters. In order to improve the sampling efficiency, PPO adopts the important sampling method to change the policy gradient algorithm from on-policy to off-policy. At this time, the update formula of the Actor network is:

$$\nabla \bar{R}_\theta = E_{\tau \sim p_{\theta_{\text{old}}}(\tau)} \left[ \frac{p_\theta(\tau)}{p_{\theta_{\text{old}}}(\tau)} R(\tau) \nabla \log p_\theta(\tau) \right] \tag{2}$$

where $\tau = \{s_1, a_1, s_2, a_2, \cdots, s_T, and a_T\}$ represents the trajectory of the agent in the entire episode. In the above formula, $p_{\theta_{\text{old}}}(\tau)$ interacts with the environment, and the parameter

of $p_\theta(\tau)$ is really trained. $r_t(\theta) = \frac{p_\theta(a_t|s_t)}{p_{\theta_{\text{old}}}(a_t|s_t)}$ indicates the update ratio of the old and new policies. The formula for updating the gradient is:

$$J(\theta) = \hat{E}_t \left[ \frac{p_\theta\left(a_t\hat{|}_t\right)}{p_{\theta_{old}}(a_t \mid s_t)} A_t \right] = \hat{E}_t\left[r_t(\theta)\hat{A}_t\right] \tag{3}$$

where $\hat{A}_t$ represents the estimation of the advantage function at time step $t$. At this time, the Actor network has been updated offline, but in order to ensure the effect of important sampling, it is necessary to limit the difference between the two distributions of the old and new policy. PPO proposes two methods to limit the update range of the policy.

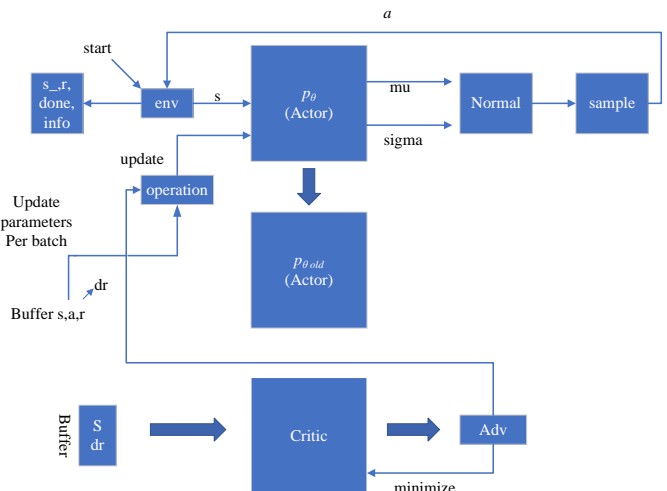

**Figure 1.** PPO algorithm training flow chart.

PPO1 uses KL divergence to measure the difference between the old and new distributions and adds it to the loss function as a penalty term.

$$J(\theta) = r_t(\theta)\hat{A}_t - \beta\text{KL}\left[p_{\theta_{\text{old}}}, p_\theta\right] \tag{4}$$

where $\beta$ is the adaptive penalty coefficient, and the KL divergence is used to measure the gap between the probability distribution of the actions output by the policy network under the same state.

PPO2 uses clip method to directly limit the update range to $[1 - \varepsilon, 1 + \varepsilon]$. The loss function of PPO2 is:

$$J(\theta) = \min\left(r_t(\theta)\hat{A}_t, \text{clip}(r_t(\theta), 1 - \epsilon, 1 + \epsilon)\hat{A}_t\right) \tag{5}$$

$$\text{clip}(x, x_{\min}, x_{\max}) = \begin{cases} x, & \text{if } x_{\min} \leqslant x \leqslant x_{\max} \\ x_{\min}, & \text{if } x < x_{\min} \\ x_{\max}, & \text{if } x_{\max} < x \end{cases} \tag{6}$$

where $\varepsilon$ is a hyperparameter that represents the maximum difference between the new policy and the old policy. The clip function limits the update range to $[1 - \varepsilon, 1 + \varepsilon]$. The PPO algorithm ensures that the policy is updated in a monotonous and undiminished direction by minimizing the loss and guarantees that the update range of the policy is controllable.

## 3. Muti-Agent Proximal Policy Optimization Algorithm

### 3.1. Algorithm Framework

In the multi-agent system, the reward obtained by each agent is not only related to its own actions but also related to other agents. Changing the policy of one agent will affect

the selection of the optimal policy of other agents, and the estimation of the value function will be inaccurate, so it will be difficult to ensure the convergence of the algorithm.

In order to solve this problem, as shown in Figure 2, this paper uses the idea of centralized training and distributed execution [25] and improves the PPO algorithm to a muti-agent proximal policy optimization algorithm (MAPPO). The Critic network learns a central value function. It can observe the global information, including other agent information and environmental information. For the Actor network, each agent only calculates the policy through its own local observation.

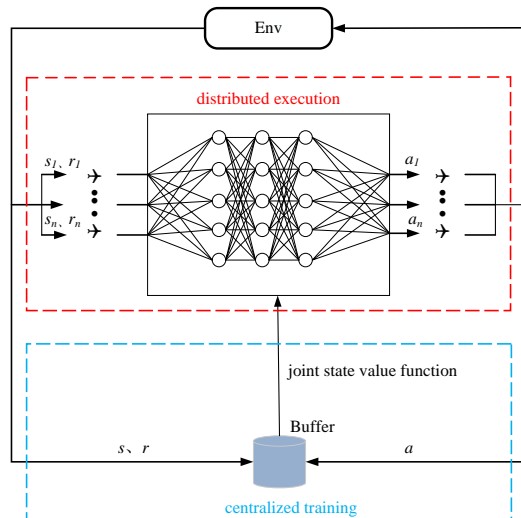

**Figure 2.** Centralized training and distributed execution.

The algorithm studied in this paper allows any number of agents to join, assuming that all agents share the same state space *S*. In the MAPPO algorithm, the Critic network of each agent fits the global value function instead of the agent's own value function. In this way, only the agent's policy needs to be updated in the direction of optimizing the global value function. Like the single-agent deterministic policy gradient, the multi-agent policy gradient can also be derived directly through the chain rule.

In a combat environment containing several UAVs, for each UAV *i*, its observation value at time *t* is $S_i$, and the Actor network outputs the mean and variance of the corresponding action probability distribution according to $S_i$, then constructs a normal distribution sampling to obtain action. Through the above methods, UAVs learn the cooperation policy between agents. In the execution phase, it only relies on its own local perception to make actions, thus realizing a collaboration policy that does not rely on communication. In addition, in order to reduce the cost of network training, UAVs with the same function share the same Actor network parameters.

### 3.2. Inherited Training Based on Course Learning

In the early stage of training, the whole UAV cluster destroys the target ship, and there are very few trajectories to complete the task. When the target ship appears randomly in the map, it will cause great interference to the learning of the agent. However, the randomization of the target ship in a certain range is more in line with the actual needs. This paper uses the idea of course learning [26] to randomize the initial coordinates of the target ship. As shown in Table 1, $x_{tar\,1}$ and $y_{tar\,1}$ represent the coordinates of the target ship 1, and $x_{tar\,2}$ and $y_{tar\,2}$ represent the coordinates of the target ship 2. The training of the whole UAV cluster is divided into three stages. First, the target ship coordinate points are fixed. Then, the initialization coordinates are randomly generated in a small area, and finally, they expand to the largest random area. In this way, as shown in Figure 3, the entire cluster will inherit the experience learned to achieve simple goals to solve more complex problems, reducing the blindness of exploration.

**Table 1.** Random range.

| Range | $x_{tar1}$/km | $y_{tar1}$/km | $x_{tar2}$/km | $y_{tar2}$/km |
|---|---|---|---|---|
| $S_0$ | 90 | 80 | 70 | 80 |
| $S_1$ | $[85, 90]$ | $[75, 80]$ | $[65, 70]$ | $[75, 80]$ |
| $S_2$ | $[85, 95]$ | $[75, 85]$ | $[65, 75]$ | $[75, 85]$ |

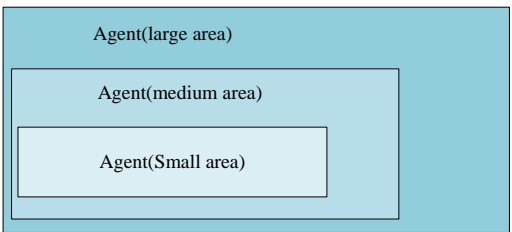

**Figure 3.** Inheritance in curriculum learning.

## 4. Cooperative Decision Model of UAV Cluster

### 4.1. Experimental Environment

In this paper, a self-developed multiple-UAV coordination system is used as the training environment.The platform is modeled on gym's reinforcement learning environment. The three-dimensional multiple-UAV coordination system is used as the training environment. The air combat environment and necessary API interfaces are redefined based on Unity3D. It can be tested by calling various classical reinforcement learning algorithms. The training environment includes 45 fighters, 5 detectors, 4 target ships, and 2 frigates. The fighters are responsible for attacking the target ship (it is believed that the fighters can destroy the target ships by hitting 10 times), and the detectors are responsible for forming a safe area to prevent the fighters from being shot down by the ship. Both the target ship and the frigate can shoot down UAVs that are not in the safe area within a certain range. There is a penalty reward for a UAV when it is shot down. The goal of the UAV cluster is for the detectors to establish a safety zone and assist the fighters to destroy the target ship with minimal casualties. The visualization results of the entire training process are shown in Figure 4.

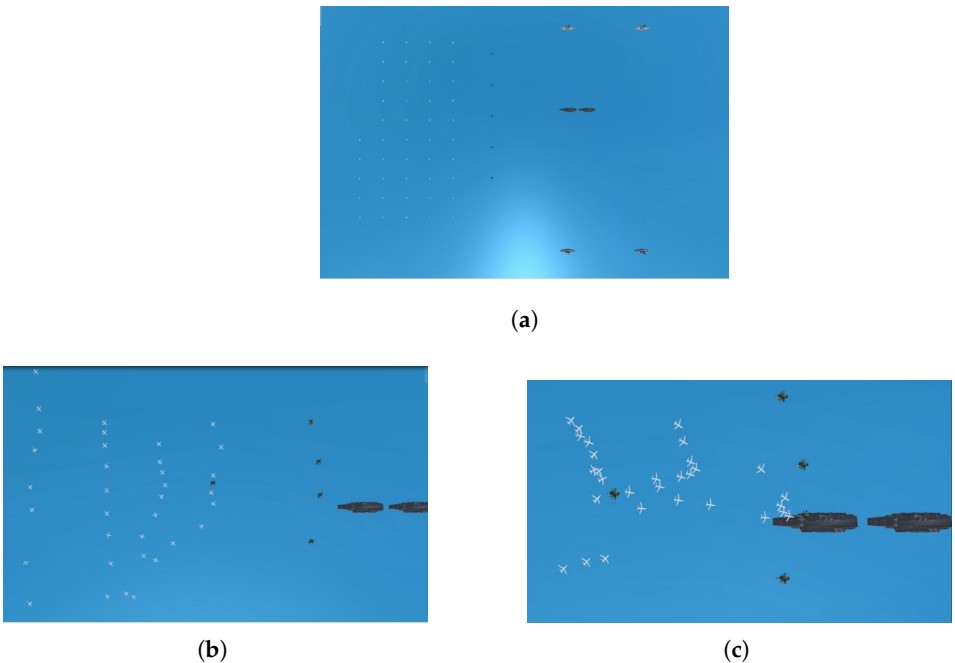

(**a**)

(**b**)                                       (**c**)

**Figure 4.** Experimental environment: (**a**) Location initialization; (**b**) Search target; (**c**) Launch strike.

In Figure 4a, after position initialization, the UAVs are driven by the parameters of the reinforcement learning model to make sequence decisions. In Figure 4b, different types of UAVs cooperate. The detectors approach the target ship to form a safe area, and the fighters approach the safe area. From Figure 4c, it can be seen that in the safe area, the fighters strike the target ship, and the heterogeneous UAV coordination policy is effective.

*4.2. Reward Mechanism*

In the task designed in this paper, the only goal of the UAVs is to destroy all target ships. At this time, the greatest reward should be given to avoid the main task being covered by other subtasks. In this paper, the entire cluster obtains a 100 reward when destroying a target ship, and a UAV obtains a −10 reward when it is knocked out by ships. At the same time, in order to prevent the sparse reward problem shown in Figure 5 [27] and improve the sampling efficiency, this paper adopts the guidance reward based on the heading angle. As shown in Equation (7), the guidance reward is calculated from the included angle between the ideal heading angle and the current actual heading angle, and the smaller the included angle, the greater the reward.

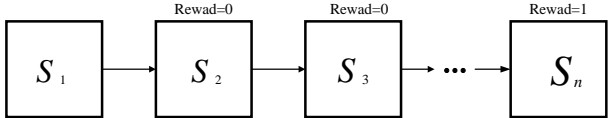

**Figure 5.** Sparse reward in reinforcement learning.

$$R_g = \pi - \left| \arctan \left( \frac{y_i - y_{\text{tar}}}{x_i - x_{\text{tar}}} \right) - \arctan(\theta) \right| \tag{7}$$

where $x_i$, and $y_i$ are the position coordinates of the *i*-th UAV, $x_{tar1}$ and $y_{tar1}$ are the position coordinates of the target ship, and $\theta$ is the heading angle of the UAV at the current time. Let $\alpha_t$ be the horizontal projection of the UAV at the time of $S_{t-1}$ to the track segment of $S_t$. Considering the limitation of the aerodynamic structure of the fuselage, there is a maximum range of the UAV's heading angle:

$$\cos(\theta) \leqslant \frac{\alpha_t^{\text{T}} \alpha_{t+1}}{\|\alpha_t\| \cdot \|\alpha_{t+1}\|}, (t = 2, 3, \cdots, n-1) \tag{8}$$

At the same time, in order to avoid collisions between UAVs, a collision penalty should be set. When there is no collision, the agent is given a small but positive reward value. When the minimum distance *l* between UAVs is less than the safe distance, it will be punished accordingly.

$$R_c = \min \left( R_{\text{pos}}, L - L_{\text{safe}} \right) \tag{9}$$

In order to maintain the normal operation of the environment and prevent UAVs from escaping the boundary during the exploration, a boundary penalty is added to the entire reward mechanism. As shown in formula (10), when the distance d of the UAV from the boundary is greater than 2 km, the boundary penalty is 0. When the distance from the UAV to the boundary is less than 2 km, the UAV is given a penalty.

$$R_d = \begin{cases} 0, d > 2 \\ 150(d-2), \text{ otherwise} \end{cases} \tag{10}$$

Considering the limitation of UAV's own performance, the speed of the UAV cannot be infinite. In three-dimensional space, the one-way speed of UAV should meet the maximum constraint:

$$|v| \leqslant v_{\text{max}} \tag{11}$$

*4.3. Network Design*

The design of the neural network is shown in Figure 6. All agents use the same neural network structure, which is a three-layer fully connected neural network with 128 neurons in each layer, and the activation function of the hidden layer is Relu. In order to improve the training speed and prevent overfitting, the hidden layers are batch normalized. Using the Adam optimizer, the learning rate is 0.005. The input of the neural network is a multidimensional state vector, which mainly includes the coordinates and survival states of all agents, whether the UAV is under the hidden state, whether the detectors interfere with the ship and so on. The output layer uses the Softmax activation function and chooses the action with the highest probability to output. In this paper, the action taken by the UAV is the speed of each dimension.

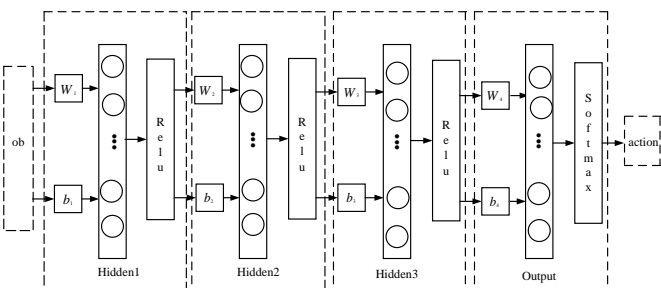

**Figure 6.** Design of neural network.

As shown in Figure 7, although the same neural network structure is used, the policy networks of the fighter and the detector are trained separately, homogeneous UAVs use the same neural network weight parameters, and heterogeneous UAVs use different weight parameters.

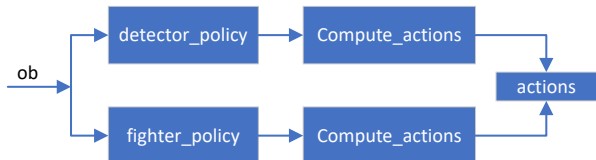

**Figure 7.** Obtaining the actions from the policy network.

## 5. Experimental Results and Analysis

*5.1. Convergence Analysis*

The experiment was conducted in a Unity3D simulation environment consisting of 45 fighters, 5 detectors, 2 target ships, and 4 frigates. The target ship had a life value of 10, and each fighter could only perform an effective strike at most. The one-way speed of the UAV in the three-dimensional space could not exceed 36 km/h, and the target ship and the frigate were traveling at a constant speed of 10 km/h in the sea. The hyperparameters of the MAPPO algorithm are shown in Table 2.

**Table 2.** Parameter configuration.

| Hyperparameters | Value |
|---|---|
| train_batch_size | 8000 |
| Gamma | 0.6 |
| epsilon | 0.2 |
| max_steps | 1000 |

In order to verify the performance of the algorithm improvement, this paper conducted a ablation experiment on MAPPO and compares its performance with the common PPO algorithm. Under the distributed Ray cluster, the model was trained for 450 epochs using 40 workers, and the accumulated reward of the two algorithms are shown in Figure 8.

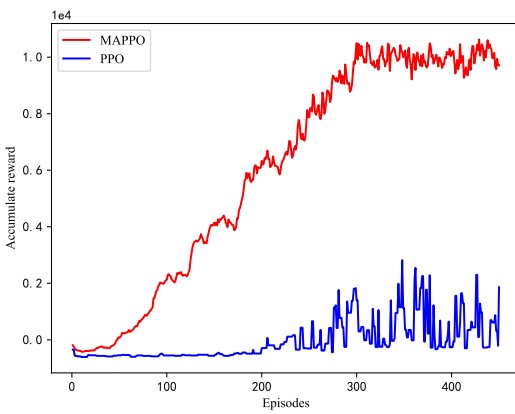

**Figure 8.** Ablation experiment.

It can be seen that the PPO algorithm cannot converge in the complex heterogeneous multiple-UAV environment. Based on the centralized training and distributed execution, the MAPPO algorithm achieves stable convergence and obtains a high average accumulated reward, which proves the effectiveness of the algorithm proposed in this paper.

We compare the MAPPO algorithm with the two mainstream multi-agent reinforcement learning algorithms COMA and BiCNet. The network structure and parameter settings of the algorithms are identical. Using 40 workers for 450 iterations, after about 6 million steps of exploration, the training curve is shown in Figure 9.

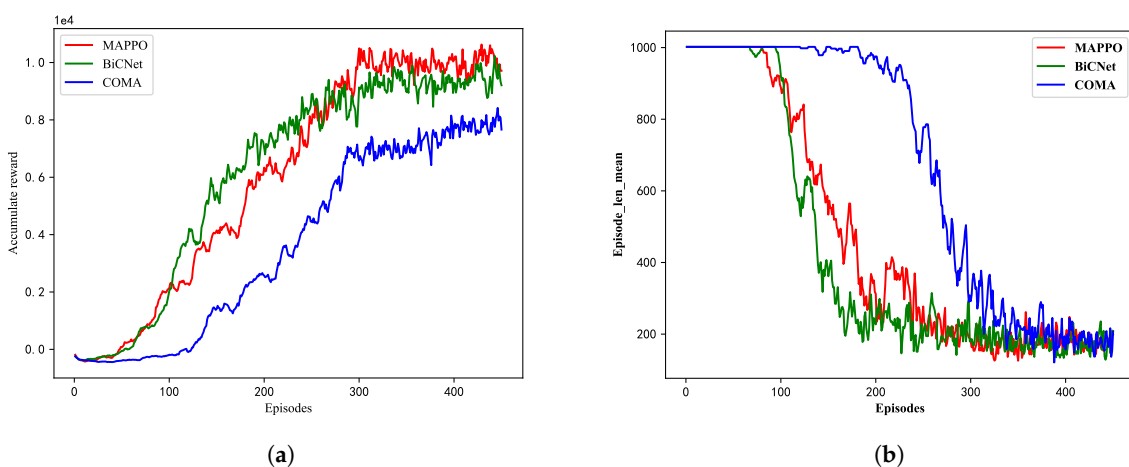

(**a**) (**b**)

**Figure 9.** The training curve of the UAV cluster: (**a**) Average accumulated reward; (**b**) The number of steps required for each episode.

It can be seen from Figure 9a that after sufficient iterations, the three algorithms all begin to converge after 300 episodes. However, the average reward value obtained by the MAPPO algorithm after convergence is significantly higher than the other two algorithms. Figure 9b shows that after the algorithm converges, MAPPO can complete the task with the minimum number of steps, which verifies the superiority of the algorithm. The statistical results of the average accumulated reward of the last 100 episodes of each algorithm and the average steps per round are shown in Table 3.

**Table 3.** The statistics of experimental results.

|  | COMA | BiCNet | MAPPO |
|---|---|---|---|
| Accumulated reward | 7571.67 | 9340.40 | 9964.01 |
| Episode_len_mean | 193.69 | 180.88 | 172.26 |

In the simulation process, the fighters and detectors were heterogeneous agents, and their functions were different. The detectors were responsible for interfering with the target ship and frigate to prevent the UAV from being detected, and the fighter was mainly responsible for carry out a suicide attack on the target ship. The average accumulated reward of the fighter and the detector for the training process are shown in Figure 10.

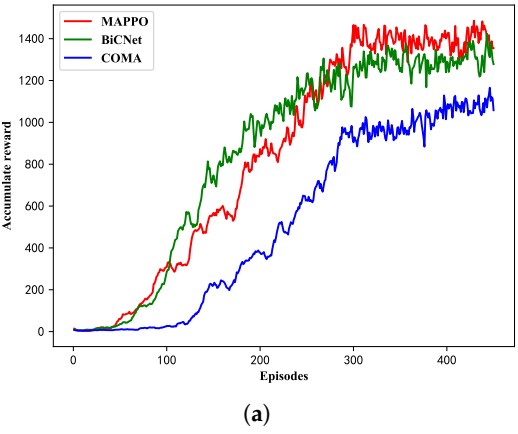

(**a**)

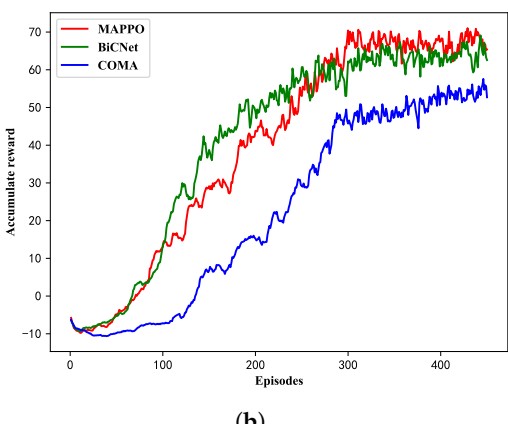

(**b**)

**Figure 10.** Average accumulated reward of detector and fighter under different algorithms: (**a**) Average accumulated reward of detector; (**b**) Average accumulated reward of fighter.

Detectors and fighters had different functions and reward function settings, so they learned two completely different polices. As can be seen from the Figure 11 (the abscissa is the number of steps, and the ordinate is the policy loss value), whether the detector's policy or fighter's policy. Its loss value shows a downward trend with the training, and finally reaches a relatively stable convergence. Driven by different policy network model parameters, detectors and fighters performed their respective duties to maximize the accumulated reward of the entire UAV cluster and finally realized multi-agent cooperation.

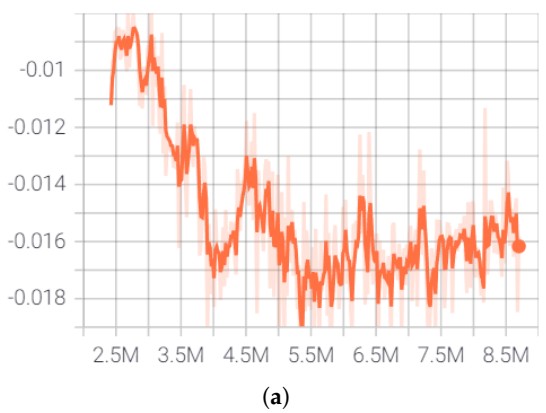

(**a**)

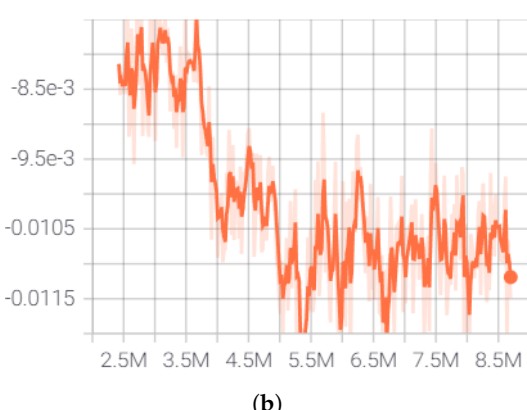

(**b**)

**Figure 11.** Policy loss curve: (**a**) Detector policy loss curve; (**b**) Fighter policy loss curve.

In order to avoid the exploration of UAVs falling into local optimization and encourage UAVs to explore more diversified actions, this paper improves the loss function of the Actor network and adds an action entropy reward. When the UAV explores a new action in the current state, it will be rewarded.

The addition of action entropy can make the policy more random, reduce the sensitivity of the algorithm to model and estimation errors, and make the algorithm more robust. As shown in Figure 12 (the abscissa is the number of steps, the ordinate is the entropy loss value), the policy entropy curves of detectors and fighters show an upward trend in the initial stage of training, indicating that the agent is constantly trying to discover new

actions and finally converge to a stable state, which fully shows the effectiveness of the improvement in this paper.

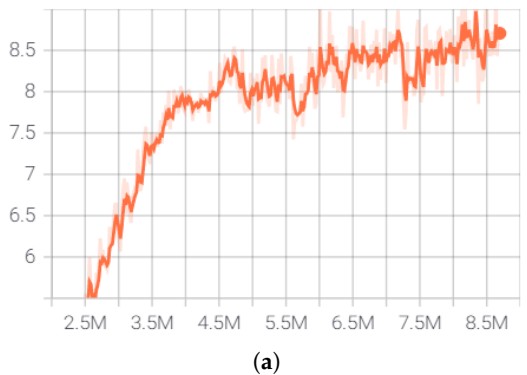
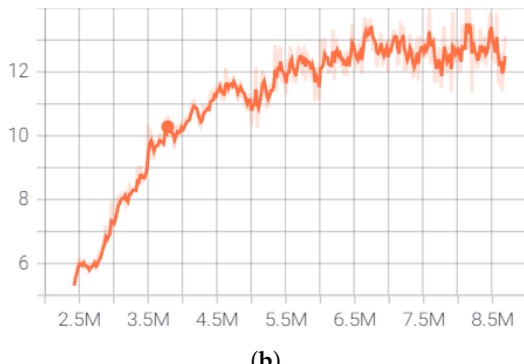

| (**a**) | (**b**) |

**Figure 12.** Policy entropy loss curve: (**a**) Detector policy entropy loss curve; (**b**) Fighter policy entropy loss curve.

*5.2. Anti-Interference Analysis*

In order to test the anti-interference performance of the algorithm, as shown in Table 4, a fixed value disturbance was added to the step function of each decision-making cycle. The win rate of 300 simulation results shows that with the increase of step disturbance, the task completion rate of the whole cluster decreases. However, even after adding an 18 m deviation to the UAV at each step, the decision model can still plan a reasonable attack route based on the current state information, and the task completion rate can remain above 82%. This means that the algorithm has a strong ability to resist environmental interference.

**Table 4.** Test results of anti-interference.

| Disturbance/m | Win Rate/% |
|:---:|:---:|
| +6 | 97.3 |
| +12 | 92.3 |
| +18 | 85.7 |
| −6 | 95.7 |
| −12 | 93.3 |
| −18 | 82.7 |

*5.3. Generalization Performance Analysis*

Generalization refers to the adaptability of the model to new samples. This paper uses the idea of course learning to achieve the task of searching and striking random target ships in a certain range, which improves the generalization performance of the decision model and meets the needs of actual combat. In order to show the generalization performance of the algorithm, outside the random range of Table 1, this paper selected four typical coordinate points of the target ship for the test. The win rate of 300 tests is shown in Table 5.

**Table 5.** Test results of anti-interference.

| Target Ship 1 | Target Ship 2 | Win Rate/% |
|:---:|:---:|:---:|
| (87, 82) | (77, 82) | 87.3 |
| (83, 80) | (63, 80) | 84.7 |
| (96, 78) | (66, 78) | 89.3 |
| (90, 86) | (80, 86) | 81.7 |

It can be seen from Table 5 that although the coordinates of the target ship are outside the maximum random range of training, with the powerful fitting ability of the neural network, the fighters and detectors can still perform their duties and make corresponding

responses according to the environmental situation to complete the target task. In summary, it can be proved that the generalization of the algorithm is guaranteed.

## 6. Conclusions

Based on the PPO algorithm and the idea of centralized training and distributed execution on the Ray framework, this paper proposes the MAPPO algorithm, which is suitable for the field of multiple UAVs, and adds the action entropy reward to the objective function to encourage diverse exploration, which is verified in a self-developed Unity3D UAV swarm combat environment. The experimental results show that the model trained based on the MAPPO can effectively complete the task, and the model has strong anti-interference and generalization, which verifies the effectiveness of the algorithm in the field of multi-agent cooperation. Next, the scope of research will be expanded to conduct research on multi-agent credit assignment.

**Author Contributions:** Conceptualization, G.Z. and X.Z.; methodology, G.Z.; software, X.Z. and Z.Y.; validation, G.Z., X.Z. and Z.L.; formal analysis, L.X.; investigation, Z.Y.; resources, G.Z.; data curation, Z.L.; writing—original draft preparation, X.Z. and D.Z.; writing—review and editing, Z.Y. and Z.L.; visualization, X.Z.; supervision, L.X.; project administration, D.Z. and Z.Y. All authors have read and agreed to the published version of the manuscript.

**Funding:** This research received no external funding.

**Institutional Review Board Statement:** Not applicable.

**Informed Consent Statement:** Not applicable.

**Data Availability Statement:** Data available on request due to restrictions eg privacy or ethical. The data presented in this study are available on request from the corresponding author. The data are not publicly available due to commercial use.

**Acknowledgments:** Not applicable.

**Conflicts of Interest:** The authors declare no conflict of interest.

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
