# Peer review of "Multiple-UAV Reinforcement Learning Algorithm Based on Improved PPO in Ray Framework"

_drones, doi:10.3390/drones6070166_

Round 1

Reviewer 1 Report

This paper takes the self-developed Unity3D collaborative combat environment as the test scenario, setting tasks requires heterogeneous unmanned aerial vehicle (UAV) to perform distributed decision-making and complete cooperation task. This paper is within the scope of Drone jouranl, the contents are novel, the results are sufficient, but it suffers several problems:

1.  In Section 1, more recent work in Drone and other journals should be further cited and discussed, such as the following papers:

Yintao Zhang and Youmin Zhang, Ziquan Yu, Path following control for UAV using deep reinforcement learning approach, Guidance, Navigation and Control, 2021, 1(1): 2150005:1-18.

Zhou Xing, Jindou Jia, Kexin Guo, Wei Jia, and Xiang Yu, Fast Active Fault-Tolerant Control for a Quadrotor UAV Against Multiple Actuator Faults, Guidance, Navigation and Control, 2022,2(1): 2250007:1-16. 

Chih-Chun Chen and Hugh H.-T. Liu, Adaptive modeling for downwash effects in multi-UAV path planning, Guidance, Navigation and Control, 2021, 1(4): 2140005:1-16. 

2. The new contents should be further highlighted with more details,  

3. The words in Figs 10 and 11 are too small, it should be redrawn

4. The English presentation is poor, it should be further polished by a native English speaker. For example, Multiple unmanned aerial vehicle (UAV) should be changed with "Multiple unmanned aerial vehicles (UAVs)"; The title is not right, Multiple UAV should be changed with "Multiple UAVs".

Reviewer 2 Report

The paper presents an extension to the PPO RL algorithm for a multi-agent settings and validates such an extension in the domain of UAV coordination.

The manuscript is well structured but poorly written, and needs to be proofread by native speaker as many sentences lack proper grammar.

There are also many little imprecisions throughout the paper, such as stating that in a single agent RL setting the change to environment only depend on agent's own actions: that's not true in general as the environment may have its own dynamics.

However, my main concerns are the following:

 - page 5 below fig 2: why "Each agent in the algorithm has a local Critic network, which receives the observations and actions of all agents. At this time, Critic learns a central value function, it can observes the global information, including other agent information and environmental information"? why replicating the critic network, that gathers global observations, in each agent? that's the opposite of what the centralised critic and decentralised execution framework is meant to do...to my understanding this is the "extension" to the PPO algorithm that the authors propose...if that's correct, I fail to understand how this solution would perform better than simply training a centralised critic...why there isn't a comparison with plain PPO?

 - ablation studies are missing, this is related to previous point: if your extension is simply to replicate the critic network in every agent, why don't you compare with plain PPO, with a single critic network?

 - the fact that the critic network is in every agent implies that every agent has to communicate all its data to any other agent....this is extremely inefficient communication-wise, yet no mention of the issue is in the paper

 - I fail to see where the cooperation is...I get that fighters are assumed to suicide into enemy ships and that detectors are meant to cover fighters, but they have fairly separated goals and authors use different weights, hence there is no evidence that cooperation actually takes place...it may be that simply agents are specialised in their role

 - the claim that PPO fails in MAS seems not well-accepted: https://arxiv.org/abs/2103.01955 

I'd consider necessary for authors to clarify the above points before considering the paper for publication

Reviewer 3 Report

The paper entitled “Multiple UAV Reinforcement Learning Algorithm Based on Improved PPO in Ray Framework” studies the distributed decision-making problem with unmanned aerial vehicles (UAVs) using reinforcement techniques. Simulation results are provided to validate the proposed methods.

However, I have some concerns as follows:

Technical aspect:

-       The literature review section at the beginning is superficial and does not represent the state-of-the-art considering different methods for UAVs distributed decision-making/coverage/path planning/formation/flocking problems. For example, some active researchers in the related disciplines are as follows:

https://scholar.google.com/citations?user=4hWyM_gAAAAJ&hl=en

https://scholar.google.com/citations?user=g-R5AZsAAAAJ&hl=en

https://scholar.google.com/citations?user=MnRdNZoAAAAJ&hl=en

https://scholar.google.com/citations?user=0h-0qjIAAAAJ&hl=en

https://scholar.google.it/citations?user=s4mZnz8AAAAJ&hl=en

-       What are the main contributions of this paper in comparison to the related state-of-the-art articles?

-       The respected authors should support the statements made in this manuscript by providing the appropriate references.

-       What is the computational complexity of the proposed method?

-       Can the proposed method be employed in real-time?

-       From practical point of view, the dynamics of the UAV is a very important part of designing a feasible area coverage. It is not clear to me how the respected authors consider the dynamics of the UAV in their proposed method?

-       I would recommend the respected authors to investigate the applicability of the proposed approach in practical systems.

-       From practical point of view, how would the proposed method perform for large-scale multi-uav systems? Would it scale well for such systems?

-       From practical point of view, what would happen if some of the uavs are removed from (or added to) the network? Does the proposed method still hold?

-       Some elements in the paper sound not clear/complete/rigorous, and sometimes vague.

-       I would recommend the respected authors to spend more time in writing the manuscript and also in highlighting the contributions of their paper.

Presentation aspect:

-       The paper needs improvement in presentation. The paper is not well written and is difficult to follow. For example, there are too many typos, and it seems that the authors did not review the paper themselves before submitting it.

-       Linguistics, readability of the paper must be improved, and the authors should restructure the paper in order to have a smooth transition among the sections.

-       Replace all blurry figures with the new ones with high quality.

-       Please make sure that all the axes are properly defined.

-       Please make sure that all plots and the axes are readable.

-       Please make sure that all the figures have same/similar format.

-       Please make sure that the figures are readable.

-       Please make sure that there is no overlap in illustrating the figures and tables.

Round 2

Reviewer 2 Report

The authors satisfied my comments and I think the additional material and modifications sensibly improves the paper.

Hence I propose to accept it

Reviewer 3 Report

N/A